# The Physiological and Biochemical Responses of European Chestnut (*Castanea sativa* L.) to Blight Fungus (*Cryphonectria parasitica* (Murill) Barr)

**DOI:** 10.3390/plants10102136

**Published:** 2021-10-08

**Authors:** Gabriella Enikő Kovács, Lóránt Szőke, Brigitta Tóth, Béla Kovács, Csaba Bojtor, Árpád Illés, László Radócz, Makoena Joyce Moloi, László Radócz

**Affiliations:** 1Institute of Plant Protection, Faculty of Agricultural and Food Sciences and Environmental Management, University of Debrecen, 138 Böszörményi St., 4032 Debrecen, Hungary; kovacs.gabriella@agr.unideb.hu (G.E.K.); radocz@agr.unideb.hu (L.R.); 2Institute of Food Science, Faculty of Agricultural and Food Sciences and Environmental Management, University of Debrecen, 138 Böszörményi St., 4032 Debrecen, Hungary; szoke.lorant@agr.unideb.hu (L.S.); kovacsb@agr.unideb.hu (B.K.); 3Institute of Land Use, Engineering and Precision Farming Technology, Faculty of Agricultural and Food Sciences and Environmental Management, University of Debrecen, 138 Böszörményi St., 4032 Debrecen, Hungary; bojtor.csaba@agr.unideb.hu (C.B.); illes.arpad@agr.unideb.hu (Á.I.); radocz.laszlo@agr.unideb.hu (L.R.J.); 4Department of Plant Sciences, Faculty of Natural and Agricultural Sciences, University of the Free State-Main Campus, P.O. Box 339, Bloemfontein 9300, South Africa; MoloiMJ@ufs.ac.za

**Keywords:** antioxidant enzyme activities, biotic stress, *Castanea sativa*, chestnut, chestnut blight (*Cryphonectria parasitica*), photosystem efficiency

## Abstract

The most important disease of European chestnut (*Castanea sativa* Mill.) is chestnut blight caused by the fungus *Cryphonectria parasitica* (Murrill) Barr which induces yield reduction in Europe and North America. This study aimed to investigate the impacts of *C. parasitica* infection on the physiological and biochemical characteristics of European chestnut at two different growth stages, 3 and 6 weeks after the infection. The amount of photosynthetic pigments (chlorophyll-*a*, chlorophyll-*b*, and carotenoids), the relative chlorophyll content, and the photochemical efficiency of the photosystem II (PSII) were measured in the leaves above and below the virulent and hypovirulent *C. parasitica* infections. The highest values were measured in the control leaves, the lowest values were in the leaves of the upper part of virulent necrosis. Antioxidant enzyme activities such as ascorbate peroxidase (APX), guaiacol peroxidase (POD), and superoxide dismutase (SOD), proline, and malondialdehyde concentrations were also investigated. In each of these measured values, the lowest level was measured in the control leaves, while the highest was in leaves infected with the virulent fungal strain. By measuring all of these stress indicator parameters the responses of chestnut to *C. parasitica* infection can be monitored and determined. The results of this study showed that the virulent strain caused more pronounced defense responses of chestnut’s defense system. The measured parameter above the infection was more exposed to the blight fungus disease relative to the leaves below the infection.

## 1. Introduction

The impact of climate change is evident across the globe. Average temperatures are rising in many parts of the world, and in parallel, weather extremes are becoming more common [1]. This is already observed because global warming has a significant impact on terrestrial biological systems [2]. The range of many animal and plant species has shifted further north [3]. The virulence of plant pathogenic microorganisms and the disease process they cause can also be significantly altered by climate change [4]. It is estimated that 20–30% of the world’s plant and animal species will be doomed to certain extinction if global temperature rises reach 1.5–2.5 °C [5]. Pests are better able to adapt to climate change relative to plants because they can change habitat and find new food resources, these are called polyphages [6], or pests can naturally hybridize to adapt to the new environmental conditions [7].

In this study, we examined the European sweet chestnut (*Castanea sativa* Mill.) belonging to the family of *Fagaceae* and one of the major fungal species damaging it, *Cryphonectria parasitica* [8]. The average temperature of 10–11.5 °C per year and the annual rainfall of 600–1600 mm is optimal for sweet chestnuts. However, due to climate change, the amount and distribution of precipitation are uncertain in most chestnut-growing areas, including Hungary, and the average annual temperature is also increasing [9]. According to the statistics of the Hungarian National Meteorological Service, the average temperature was 1.1 °C higher in 2020 than the average measured in the period 1981–2010 [10]. The distribution of precipitation is extreme, as a fraction of the average amount fell during the spring period. On average, we had the 4th driest spring in the last 120 years, while in June there was more than 50% more rain in 2020 [11].

Chestnut blight caused by *Cryphonectria parasitica* [12] attacks *Fagaceae*, including sweet chestnuts, as a biotic stress factor it causes adverse changes in the plant’s transport tissues [13,14], secretes toxins and oxalic acid that break down plant cell walls and generate cankers on tree bark [15,16,17]. They cause fundamental damages to the physiological processes of the plant by partially or completely drying the branches above the necrosis [18]. They can also affect the function and activity of antioxidant enzymes [16]. The necrosis fungus, similar to chestnut blight, attacks the transport tissues of the tree’s bark at temperatures ranging between 18 and 27 °C [19]. The germination optimum of *C. parasitica* spores is 21 °C [20], while the ejection of ascospores is between 15 and 25 °C [21].

The fungus has 74 EU vegetative compatibility strains in Europe [22]. In our earlier experiment, the growth of *Cryphonectria parasitica* was measured on six different types of media and diverse temperatures (from 5 °C to 35 °C) [23]. It can be assumed, that some strains are more vigorous than others [24] and the virulence intensity depends on the weather conditions too, so climate change can be a fundamental factor. Chestnut trees may die in a few years due to a high virulent strain and only some necrosis may occur on branches due to others.

The primary defense system of plants is usually their cuticle, but when pathogens breach this barrier, they react to the damage with some physiological change, such as the production of various compounds, possibly decreasing or increasing the production of some enzymes, and their quantitative change is well-measurable [25]. Furthermore, the reactive oxygen species (ROS) in plants, despite their destructive effects, are secondary indicators [26]. When plants produce ROS moderately, they may indicate stress tolerance, e.g., induced hydrogen peroxide improved the drought tolerance responses in soybeans [27]. Under oxidative stress conditions caused by biotic stressors, the activity of antioxidant enzymes, including superoxide dismutase (SOD), guaiacol peroxidase (GPX), and ascorbate peroxidase (APX), are increasing [25,28]. Proline is one of the basic molecules that indicates oxidative stress conditions [29]. Under the influence of biotic stress, its level initially decreases sharply and then begins to rise [30]. 

Szőke et al. [31] investigated the effect of corn smut (*Ustilago maydis*) infection. Their results showed that some physiological processes in maize (antioxidant enzyme activity) were significantly influenced by the presence of the fungus. In this experiment, we also investigated the impacts of other kinds of fungus, namely chestnut blight fungus. Whether the change in some enzyme activity in the leaves above (F) and below (A) the infection can be measured in the case of sweet chestnut plants artificially infected with bark cancer fungus even before the fungal infection causes visible symptoms. The *C. parasitica* fungus has two basic variants in terms of pathogenicity: virulent and hypovirulent types. In the case of infection caused by the hypovirulent strain, only superficial necrosis occurs, which quickly becomes callus [32]. This is due to the decreased virulence of the hypovirulent isolate [33,34]. The phenomenon of hypovirulence is caused by the modification of the physiological function of the pathogen due to the mycovirus found in the cytoplasm of the fungus [35]. This double-stranded RNA-containing virus without coat protein [36] is *Cryphonectria hypovirus* (*CHV1*). Field biological control with hypovirulent fungal strains is very common, especially on the European continent, where this method has proven to be effective in controlling the pathogen [37,38].

## 2. Results

There were no significant differences in relative chlorophyll content during the first measurement in the leaves below the infection (A). However, the values measured in the leaves located above the infection (F) were significantly lower compared with the non-infected control (Control F). At the time of the second measurement, the difference between the leaves from the control and infected branches was visible. The measured SPAD values in both the hypovirulent and virulent fungus-infected branches were lower compared with the first measurement (Figure 1).

During the first sampling, the value of chlorophyll-a in the smaller control leaves (Control-A) developed similarly as in the leaves above the branch infected with the virulent fungus (Virulent-F), but this similarity no longer existed for the second sampling. At this point, the value of chlorophyll in control leaves (Control-A, Control-F) is already three times as high as near necrosis. That is, infection (especially as the disease process progressed) had a very negative effect on chlorophyll synthesis (Figure 1). In the second measurement, chlorophyll-a was significantly high for the leaves below hypovirulent fungal strain (Hypovirulent A) treatment than the leaves in the same position on the branch treated with the virulent strain (Virulent A).

Chlorophyll-b followed the same pattern as chlorophyll-a. The highest values were measured in the control leaves, followed by the leaves above the hypovirulent and virulent lesions (Figure 1). In the second measurement, the value for chlorophyll-b around the necrosis was statistically lower than in the control leaves, and there was no significant difference in the values measured in the leaves below and above the necrosis.

The carotenoids concentration did not show any spectacular differences between the samples during the first sampling; however, the carotenoids concentration measured in the control leaves (Control-A, Control-F) during the second sampling was significantly higher than in the leaves below the necrosis (Hypovirulent-A, Virulent-A). Lower values were observed in the leaves above necrosis (Hypovirulent-F, Virulent-F) (Figure 1).

In fluorometer measurements, there were no significant differences in F_m_/F_v_ values during the first sampling; however, we measured the highest values in adult control leaves (Control-F), followed by leaves of hypovirulent fungus-infected parts, and the lowest fluorescence was measured among the leaves of a branch infected with virulent fungi (Figure 2). 

The destructive effect of the virulent fungal strain was already observed in the F_v_/F_o_ values, as here we measured a statistically lower quotient in the leaves than in the leaves in the branch damaged by the hypovirulent fungus, as well as in the control (Control-F) leaves. The trend was similar in the second sampling, but in the case of lowest fluorescence was measured in the leaves from the branches infected by the hypovirulent strain (Hypovirulent-A, Hypovirulent-F) (Figure 2).

Plant responses to biotic stress in terms of SOD values are well-shown in Figure 3. While the values measured in the leaves above the necrosis (Hypovirulent-F, Virulent-F) decreased, in the case of those below (Hypovirulent-A, Virulent-A) there was an increasing tendency in the first measurement to the control leaves (Control-A, Control-F) compared. However, in the case of the second sampling, the SOD values around the necrosis following the stress response were statistically significantly higher than in the case of control (Control-F, Control-A) leaves.

The results of lipid peroxidation measurements at the time of the first sampling were also statistically higher in the leaves above hypovirulent (Hypovirulent-F) and virulent (Virulent-F) necrosis than in the control leaves. The amount of malondialdehyde (MDA) measured in leaves under artificial infection (Hypovirulent-A, Virulent-A) was similar to that in control leaves (Control-A). These differences changed somewhat during the second sampling, however, the values of the leaves above the infections (Hypovirulent-F, Virulent-F) were also statistically different from the others but were similar to each other (Figure 4).

Similar to the data obtained for MDA, the measurements of guaiacol peroxidase (POD) at the first sampling in leaves above the branch (infected with both types of fungi) (Hypovirulent-F, Virulent-F) were statistically higher compared with the other results. However, they also differed statistically from each other and the control (Control-F, Control-A) plants (Figure 5).

At the time of the second sampling, the values measured in the leaves above necrosis (Hypovirulent-F, Virulent-F) were statistically different from the values measured in the other leaves and they also differed from each other. Plant physiological changes caused by virulent fungal infection are well-measurable (Virulent-F was highest). At that time, the POD activity was higher (5% significance level) even in the leaves under necrosis (Virulent-A) than in the leaves of the branches infected with the hypovirulent fungal strain (Hypovirulent-A).

The results of ascorbate peroxidase (APX) activity also show that the values measured in the leaves above the necrosis (Hypovirulent-F, Virulent-F) were statistically higher in the control (Control-A and Control-F) and below the necrosis (Hypovirulent-A), compared with values measured in Virulent-A leaves. Furthermore, the values of APX measured in leaves above virulent injury (Virulent-F) and leaves of branches infected with hypovirulent fungi (Hypovirulent-F) also differed. Due to virulent fungal infection, this value was 30% higher compared with hypovirulent infection. These differences were similar in the second sampling (Figure 6).

At the time of the first sampling, the leaves of the control plants had higher proline levels than the other fungal-infected plants. However, in the case of the second sampling, the leaves above the lesion (Virulent-F) of the plants treated with the virulent strain had the highest proline content, which was also statistically different from the other values. In addition, all measured values were higher compared with the first sampling (Figure 7).

## 3. Discussion

Most research has focused primarily on plant responses to abiotic stress [27,39,40,41,42], but there is also an increasing number of studies that also examine some form of biotic stress factors [11,43,44,45]. The crops included in the experiments come primarily from the most important crops, such as maize. According to several studies, plant antioxidant enzyme processes are significantly influenced by biotic stressors, and this can be well-measured by changes in the production of antioxidant enzymes [46,47]. The sweet chestnut is not as economically important as corn or wheat. Its products can also be consumed by diabetics and gluten-sensitive people, so its importance has greatly increased in recent decades and is expected to increase further [48]. This study observed the extent to which enzymatic responses are altered by biotic stress under the influence of one of the most frequently attacking fungi, *Cryphonectria parasitica*.

During the evaluation after measuring the SPAD values, it was already seen that there was a significant negative difference in the chlorophyll concentrations of leaves developed on branches infected with fungi (hypovirulent and virulent type) compared with healthy leaves (control). At the time of the second measurement, similar results were obtained, with a clear difference between the SPAD values of healthy leaves and leaves developing on fungal-infected branches. Szőke et al. [49] also measured lower SPAD values in maize plants infected with smut (*Ustilago maydis*) compared with control plants, as did Frommer et al. [50].

At the first sampling time, the values of chlorophyll-a in the control leaves were higher than in the leaves of the branch parts damaged by the virulent fungal strain with strong destructive properties. However, stress due to infection by the hypovirulent fungal strain also reduced chlorophyll levels by nearly 80% compared with Control-F leaves. This experiment showed a more drastic change than the studies of smut in a corn plant by Horst et al. [51], where the reduction in chlorophyll was only around 10% due to fungal infection. Chlorophyll-b values developed similarly in control and infected leaves. Lobato et al. [52] examined the impacts of *Colletotrichum lindemuthianum* infection on common beans’ photosynthetic pigment content. They found that the concentration of carotenoid was 28.3% and 35% lower in the infected beans’ leaves relative to the non-infected ones 8 and 12 days after the infection. Additionally, they found a correlation between the total chlorophyll level and the rate of photosynthesis. In this study, the carotenoids concentration did not change significantly in the leaves located above the infection. While the measured values were significantly higher in the leaves below the chestnut blight infection 3 weeks after the infection. The carotenoid concentration significantly decreased by 61.56% and 58% in the leaves from the branches infected by hypovirulent and virulent strains below the infection, 57% and 50% above the infection (Figure 1).

Biotic stressors also have a negative effect on photosystem II (PSII) processes. Several studies have been conducted in this regard, such as the interactions between oats and crowned rust (*Puccinia coronata*) in the 1996 [53] experiments of Scholes and Rolfe. Their results showed a decrease in F_v_/F_m_ values in infected plants, as in our studies in sweet chestnuts due to bark fungal infection. This decrease was also statistically measurable. Similar studies have been performed on oaks (*Quercus petraea*) belonging also to the *Fagaceae* family. Repka [54] measured the effect of powdery mildew (*Erysiphe cichoraceum*) on PSII processes in oak leaves. He found that the fluorescence was significantly reduced by the presence of the fungus. We came to a similar result in this study. In healthy plants, the F_v_/F_m_ value was almost 10% higher in the first measurement than in the leaves measured above virulent necrosis (Virulent-F), while it was already 16% higher in the second measurement.

Savaci et al. [55] performed similar studies on sweet chestnuts. Chlorophyll-a and -b, as well as carotenoid values, protein, the activity of superoxide dismutase (SOD) and ascorbate peroxidase (APX), were measured in the leaves of old chestnut trees, as well as in the leaves of middle-aged and young trees and the leaves of bark-infected trees. The results of Savaci et al. [55] showed that carotenoid content, APX, and SOD values were lower in infected trees than in healthy ones, while their protein content was higher. In this experiment, we demonstrated that SOD values were lower in leaves from infected branches than in healthy ones during both the first and second sampling. However, this difference was not significant in all cases. In the first sampling, the superoxide dismutase (SOD) values measured in the smaller control leaves (Control-A) were similar to those measured in the leaves above the infected branches (Hypovirulent-F, Virulent-F) with measured data of leaves developing under infected branches (Hypovirulent-A, Virulent-A). This may be since the measurements were performed on day 35 after budburst, when the plant fluid flowed intensively from the roots into the upper part of the plants, suggesting that the superoxide dismutase enzyme was affected by the presence of the fungus, therefore, there were significant differences in the measured SOD values in the leaves above (Virulent-F, Hypovirulent-F) and below the lesions (Virulent-A, Hypovirulent-A). Savaci et al. [55] collected their examined leaf samples in August, so about 80–100 days after budburst, which may be the reason for the substantial deviation of our results from theirs. At the time of the second measurement, we already found a clear difference between the control, hypovirulent and virulent fungus-infected plants. The results of chlorophyll-a, chlorophyll-b, and carotenoids content measured were significantly higher in the case of larger leaves of control plants (Control-F). In this case, too, we hypothesize that we obtained different results than Savaci et al. [55] due to the intense root water and nutrient flow during initial plant development.

The activity of ascorbate peroxidase developed similarly to the studies of Savaci et al. [55]. In the leaves growing on the branch infected with the potent necrotizing virulent strain, higher values were obtained above and below the lesion (at a significance level of 5%) during the second sampling.

Changes in proline levels in plants indicate abiotic and biotic stressors [56]. Thus, a biotic factor may initially decrease proline levels in plants [20], as confirmed by the current study. At the time of the second measurement, however, a verifiable difference was obtained between the leaves of the control and infected branches. A 30% higher proline level was measured in the leaves of the branches damaged by the virulent fungus.

Based on the results, we state that both the virulent and hypovirulent versions of the fungal species *Cryphonectria parasitica* infecting sweet chestnuts cause significant physiological changes in the host plant and the activity of antioxidant enzymes such as SOD and POD. This can be well-detected even with simpler tests (such as relative chlorophyll and photosynthetic pigments content measurements) even in the very early stages of the fungal infection. In conclusion, this experiment proved that the virulent and the hypovirulent strains of *C. parasitica* infection caused different defense responses of sweet chestnut. The results of this experiment proved our hypothesis that the responses of leaves above the chestnut blight infection are more pronounced relative to the leaves below the infection. In addition, the more virulent strain caused more severe damage to the sweet chestnut (based on measuring the amount of photosynthetic pigments, the relative chlorophyll content, and the photochemical efficiency of the photosystem II), and that this more virulent strain elicited a more intensive defense reaction (indicated by measuring of antioxidant enzyme activities, and concentrations of proline and malondialdehyde). This means that the strong virulence of that strain is not caused by the down-regulation of plant defense mechanisms.

## 4. Materials and Methods

### 4.1. Experimental Conditions and Treatments

This experiment was set up by infecting chestnut branches with both pathogenicity types of the fungus (virulent and hypovirulent). This study aimed to examine the defense responses of sweet chestnut to a virulent and a hypovirulent strain of *C. parasitica* infection. This study hypothesized that the responses of chestnut to the examined infection are more pronounced in the leaves above the *C. parasitica* infection because the part of the tree above the canker starts to die after the infection. The samples were taken from the leaf located below and above the infection, 3 weeks (21 days) and 6 weeks (42 days) after the artificial infection. The sampling times were determined based on our previous fungal growth studies on the medium [36] and Guérin and Robin’s ascospore scattering experiment [21]. 

The experiment was carried out on 5-year-old Hungarian (Nagymarosi 22) sweet chestnut (*Castanea sativa* Mill.) samplings from a South-Transdanubian breeding farm. Virulent and hypovirulent isolates of *Cryphonectria parasitica* from the strain collection of the Institute of Plant Protection of the University of Debrecen were used for artificial infection. The virulent strain is from Tiszafüred, a highly developed fungus, belonging to the EU-2 vegetative compatibility group, while the hypovirulent strain was from a Nagymaros district, also from this vegetative comparative group.

These were propagated on a Potato Dextrose Agar medium and placed under the bark in an approximately 0.5 cm × 0.5 cm mycelium-medium cube sealed with parafilm. In each case, the infection was carried out on branches approximately one centimeter in diameter to ensure necrotization of the spleen and cambium. Our previous experiments revealed [57] that the death of the bark on the thin branches was evident 4–5 weeks after infection and leaf drying symptoms occur 6–8 weeks post-infection. The budburst occurred after a short warmer period, followed by an artificial infection on day 14 (6 May 2021), and sampling on days 35 and 57 after budburst. The first sampling was on 27 May 2021, the 3rd week after the artificial infection. The time of the second sampling was 16 June 2021, 6 weeks after infection.

The experiment was carried out under non-irrigated conditions. The weather and the soil condition database measured and collected by the Institute of Plant Protection, University of Debrecen’s experimental site (47°33′07.7″ N 21°36′00.3″ E). April was cool, barely reaching an average temperature of 10 °C by the middle of the month (Figure 8), which was not suitable for either sweet chestnut or fungal development. Therefore, no artificial infection occurred at this time. At the beginning of May, the average temperature was between 10 and 15 °C, with a maximum of 15–25 °C, which is more optimal for the plant and the fungus that infects it [22].

The amount of natural precipitation in the 2 weeks before infection was around 20 mm. This amount corresponds to the amount of average annual precipitation [58]. No exceptionally large amounts of precipitation fell during the entire experiment, so this did not affect the development of our results. After the first sampling, a wetter period followed, but even then, no more rain fell than average (Figure 8).

At the time of the first sampling (27 May 2021), symptoms indicating the presence of the fungus, such as leaf dehydration, were not yet visible near the artificial lesion, even the typical orange stroma of the virulent fungus and the characteristic brown bark tissue necrosis did not appear. Control-A leaf samples from the control plant were smaller, typically weighing less than 0.5 g and 8–10 cm long, 3–4 cm wide, and were collected from the lower third of the shoot. While the Control-F samples were between 0.5 g and 0.8 g, as well as 13–14 cm long and 5–6 cm wide, taken from the middle third of the seedling. In the period of the second sampling (18 June 2021), the leaf samples had similar parameters. Even at this time, we did not observe any leaf-drying symptoms suggestive of fungal exposure; however, they were already conspicuous in the week of the following sampling, so they occurred at week 7 after artificial infection. 

### 4.2. Quantity of the Photosynthetic Pigments

The relative chlorophyll content of the leaf was measured with a SPAD-502+ Chlorophyll Meter (Minolta, Japan). The concentration of photosynthetic pigments was measured based on Moran and Porath [59] and calculated according to Welburn [60]. A fifty mg fresh leaf tissue sample was taken dissolved in 5 mL of N, N-dimethylformamide at 4 °C for 72 h. During the measurements, the absorbance at 480 nm, 647 nm, and 664 nm was measured at three wavelengths using a spectrophotometer (Nicolet Evolution 300 UV-Vis Spectrometer).

### 4.3. Measurement of Photochemical Efficiency

The photosystem II (PSII) photochemical efficiency was determined using the pulse modulated chlorophyll fluorescence induction method [61,62] with an OS5p+ pulse-modulated portable chlorophyll fluorimeter (Opti-Sciences, Hudson, USA). Five leaves from each treatment (control, virulent and hypovirulent fungal strains) were used to measure the photochemical efficiency. The leaves were dark-adapted for 20 min before the measurements, to detect the minimum chlorophyll-*a* fluorescence (F_o_). The measurements were taken at 10:00 a.m. at both sampling times. 

### 4.4. Enzyme Assays

Samples (10-10 leaves, below and above the artificial infection) were preserved in liquid nitrogen at the time of collection and stored at −80 °C until processed. The method developed by Beyer and Fridovich [63] was used to measure superoxide dismutase (SOD) activity. A frozen leaf sample (0.4 g) was pulverized in liquid nitrogen to a fine powder. Thereafter, 4 mL of buffer solution was added. This buffer contained 2 mL of 50 mM phosphate buffer (pH 7.8) and 0.1 mM EDTA, 1% (*w*/*v*) polyvinylpyrrolidone (PVP) and 1 mM phenylmethanesulfonyl fluoride (PMSF). The samples were then centrifuged at 10,000× *g* for 15 min. One unit of SOD was defined as the amount of enzyme required to cause 50% inhibition of the reduction in NBT as monitored at 560 nm.

The method developed by Zieslin and Ben-Zaken [64] was followed for guaiacol peroxidase (POD) activity. The reaction mixture consisted of 50 µL 0.2 mM hydrogen peroxide, 100 µL 50 mM guaiacol, 340 µL distilled water, 500 µL 80 mM phosphate buffer (pH 5.5), and 10 µL enzyme. An increase in absorbance as a result of tetraguaiacol formation was measured at 470 nm for 3 min at 30 °C. The blank did not contain the enzyme. An extinction coefficient of 26.6 mM^−1^ cm^−1^ was used to calculate the enzyme activity.

A method described by Mishra et al. [65] was modified for the determination of ascorbate peroxidase (APX) because no specific literature was available for the measurement of this enzyme from sweet chestnut leaves. The modifications were as follows: the assay mixture (1 mL) consisted of 520 µL of 50 mM potassium phosphate buffer (pH 7.0), 200 µL hydrogen peroxide (0.1 mM), 180 µL sodium ascorbate (0.5 mM), 50 µL ethylenediamine-tetraacetic acid (0.1 mM EDTA), and 50 µL enzyme extract. The reaction was initiated by adding the enzyme extract. A decrease in absorbance as a result of ascorbate oxidation was measured at 290 nm for 5 min at 20 °C against a blank in which the enzyme was replaced with phosphate buffer. An extinction coefficient of 2.8 mM^−1^ cm^−1^ was used.

### 4.5. Rate of Lipid Peroxidation

The method used to measure the rate of lipid peroxidation was developed by Heath and Packer [66]. The liquid nitrogen powdered leaf tissues (0.1 g) were homogenized in 1 mL 0.25% (*w*/*v*) thiobarbituric acid (TBA) and 10% (*w*/*v*) trichloroacetic acid (TCA) was added. The samples were centrifuged at 10,800× *g* for 25 min at 4 °C. Supernatants (0.2 mL) were used and added to 0.8 mL 20% (*w*/*v*) TCA and 0.5% (*w*/*v*) TBA (into a clean Eppendorf tube). The mixture was vortexed and incubated at 95 °C for 30 min, followed by cooling on ice and centrifuged once at 10,800× *g* for 10 min at 4 °C. The absorbance was measured at 532 and 600 nm. The amount of malondialdehyde (MDA) was calculated with the use of an extinction coefficient of 155 mM^−1^ cm^−1^.

### 4.6. Proline Determination

The method of Carillo and Gibon [67] was followed with some modifications. Liquid nitrogen powdered leaf tissue (0.3 g) was homogenized in 6 mL 70% (*v*/*v*) ethanol [68]. One mL ninhydrin (1% ninhydrin (*w*/*v*) in 60% (*v*/*v*) acetic acid) was added to 500 μL ethanolic extract (in 1.5 mL Eppendorf tube). The samples were incubated at 95 °C for 20 min, cooled off, and centrifuged at 12,000× *g* for 1 min. The absorbance of samples was measured at 520 nm, and the amount of proline was calculated from the proline standard curve.

### 4.7. Statistical Analyses

The statistical evaluation of the results was performed with the IBM SPSS 25 software. The normality of the data was determined by the Shapiro–Wilk and Kolmogorov–Smirnov tests. The results were then compared using the Tukey-HSD test. Five samples per treatment and per parameter were used for the statistical evaluation.

## Figures and Tables

**Figure 1 plants-10-02136-f001:**
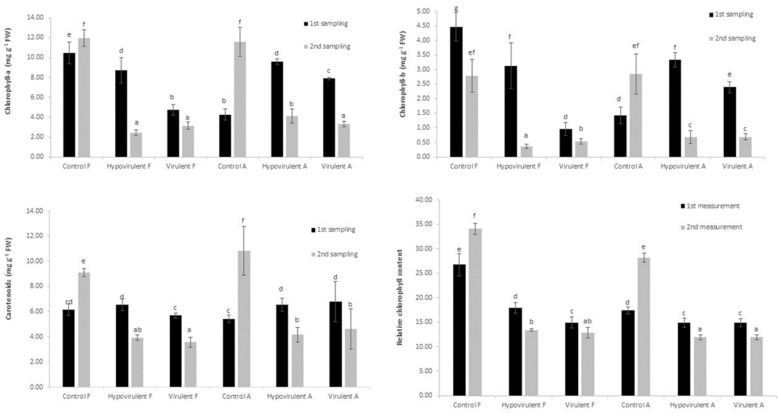
Chlorophyll-a, chlorophyll-b, carotenoids, and relative chlorophyll content (SPAD-Units) after the artificial infection (first measurement after 21 days from infection, second measurement after 42 days). Values are means ± SD, n = 25. Significant differences among samples based on the Tukey-HSD test. F: leaves above the infection, A: leaves below the infection.

**Figure 2 plants-10-02136-f002:**
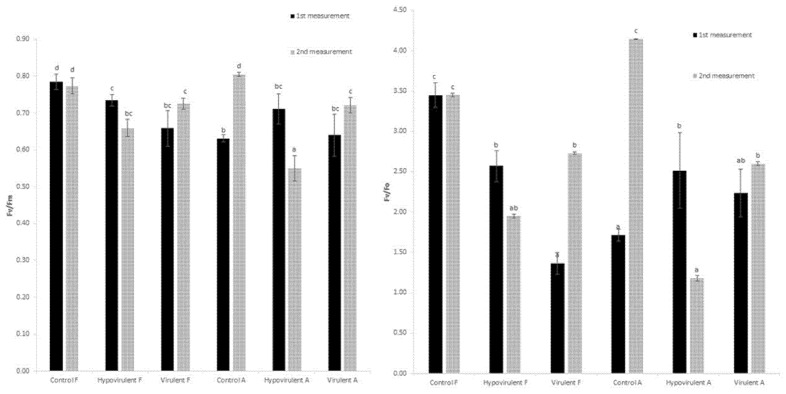
Measured chlorophyll-*a* fluorescence (variable fluorescence/maximum fluorescence) at 21 and 42 days after inoculation with hypovirulent and virulent fungal strains. Values are means ± SD, n = 5. Significant differences among samples based on the Tukey-HSD test. F: leaf above the infection, A: leaf under the infection.

**Figure 3 plants-10-02136-f003:**
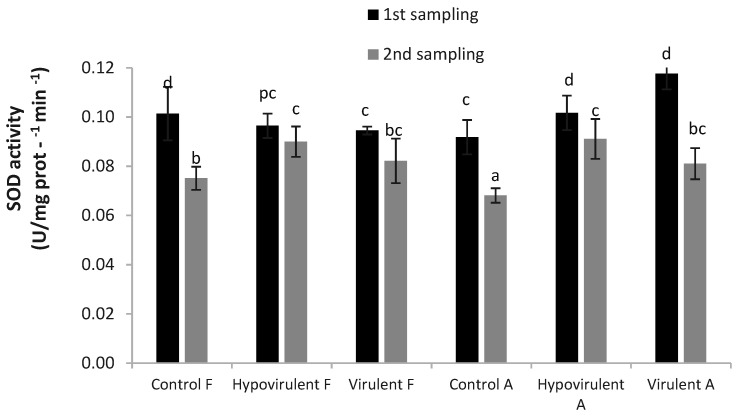
The effect of *Cryphonectria parasitica* hypovirulent and virulent strains on SOD activity of *Castanea sativa*, 21 and 42 days after infection. Values are means ± SD, n = 5. Significant differences among samples based on the Tukey-HSD test. F: leaf above the infection, A: leaf under the infection.

**Figure 4 plants-10-02136-f004:**
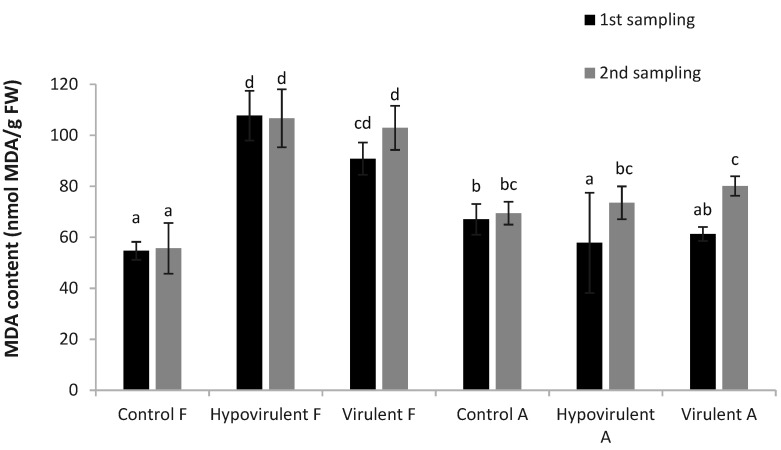
The malondialdehyde (MDA) concentration after infection (21 and 42 days). Values are means ± SD, n = 5. Significant differences among samples based on the Tukey-HSD test.

**Figure 5 plants-10-02136-f005:**
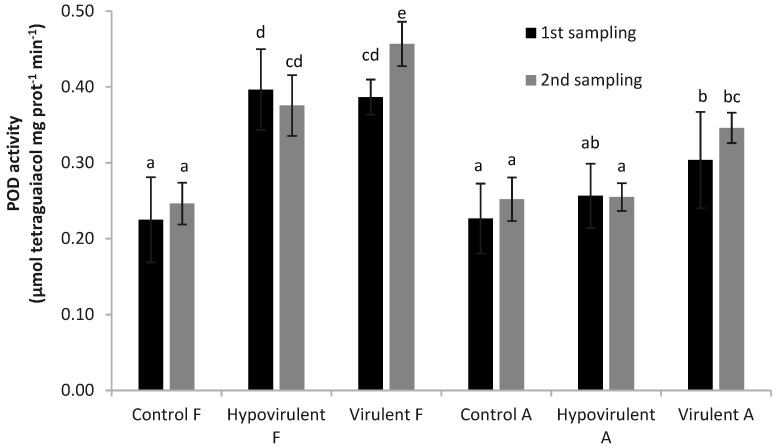
POD activity in chestnut leaves 21 and 42 days after *Cryphonectria parasitica* infection. Values are means ± SD, n = 5. Significant differences among samples based on the Tukey-HSD test.

**Figure 6 plants-10-02136-f006:**
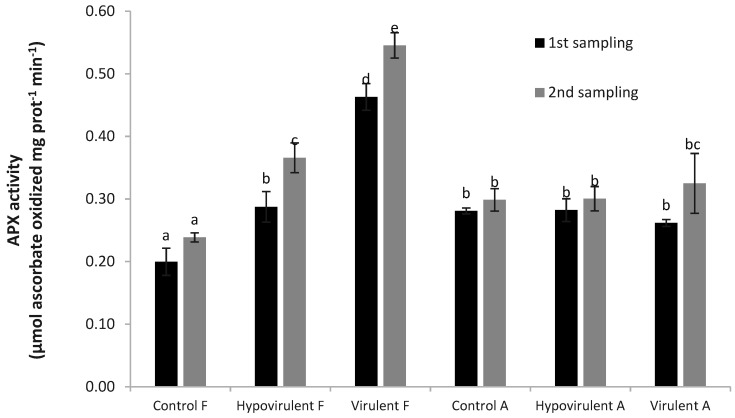
The activity of ascorbate peroxidase 21 and 42 days after the infection. Values are means ± SD, n = 5. Significant differences among samples based on the Tukey-HSD test.

**Figure 7 plants-10-02136-f007:**
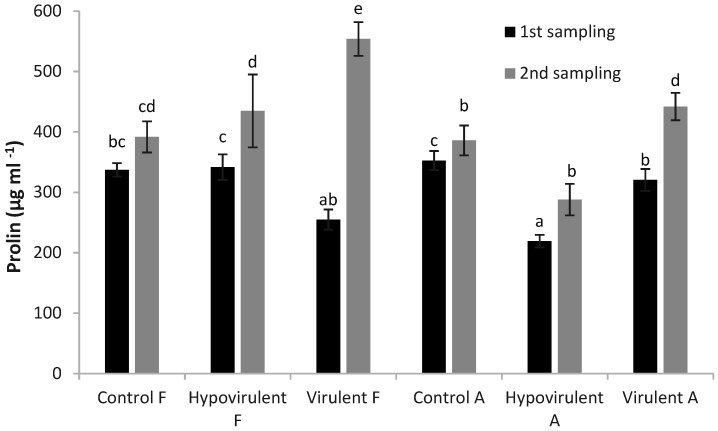
Proline content in leaves 21 and 42 days after infection. Values are means ± SD, n = 5. Significant differences among samples based on the Tukey-HSD test.

**Figure 8 plants-10-02136-f008:**
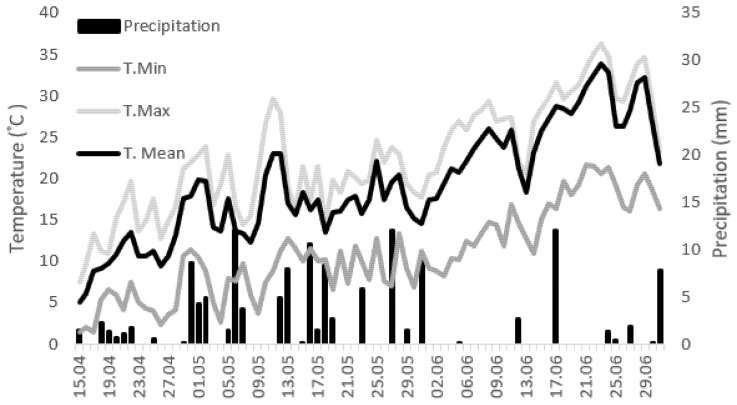
Temperature changes and precipitation in the experimental area from 15 April 2021 to the 2 weeks following the last sampling. The data originate from the Institute of Plant Protection, University of Debrecen’s meteorological station (47°33′07.7″ N 21°36′00.3″ E).

## Data Availability

The data presented in this study are available upon request from the first author.

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
