# Peer review of "The Physiological and Biochemical Responses of European Chestnut (Castanea sativa L.) to Blight Fungus (Cryphonectria parasitica (Murill) Barr)"

_plants, 2021, doi:10.3390/plants10102136_

Round 1
Reviewer 1 Report
The study compares the effects of two strains of one pathogenic fungus differing in the level of virulence on its host plant, a chestnut. It found, that the more virulent strain caused more severe damage to the plant (based on measuring of the amount of photosynthetic pigments, the relative chlorophyll content, and the photochemical efficiency of the photosystem II), and that this more virulent strain elicited more intensive defence reaction (indicated by measuring of antioxidant enzyme activities, and concentrations of proline and malondialdehyde). It means that the strong virulence of that strain is not caused by downregulation of plant defence mechanisms (at least not that, which vere studied) what is interesting and of more general interest. Unfortunately, such an interpretation is not clearly presented nor discussed by the authors, which gratuitously reduces the value of the manuscript.
The biggest weakness of the article is the quality of the English, especially in terms of phrasing. The article needs to be rewritten to make it more clear, readable and to avoid possible misleadings. I was unable to find any way to download the submitted manuscript in some editable format (e.g. docx) and I got only a pdf file which is hard to correct thoroughly. Thus, I am not submitting a list of specific corrections.
The data and results seem to be accurate and interesting, so I will keep my fingers crossed to the authors to successfully rephrase the text and to prepare the new text as carefully as the data.
Jan Ponert
Author Response
The authors wish to thank the reviewer for the valuable comments. The authors are very sorry that the reviewer was not able to download the doc format. The authors uploaded their manuscript as a doc file and MDPI converted it into pdf as a review format. So, the doc format is available so maybe the reviewer should ask MDPI about the doc format.
The authors also used the doc file format in the revised version. The authors regret this format misunderstanding and that they did not receive detailed corrections because of it. We tried our best and corrected the manuscript based on the reviewer’s suggestions. If the reviewer has any additional specific correction suggestions, please let the authors know.
The manuscript is checked by a native English speaker so hopefully, the manuscript is more understandable now.
Reviewer 2 Report
The manuscript raises many interesting issues, and the experiment was designed correctly. The results of the experiments support the conclusions drawn.
My main concerns relate to the visualization of work. Monotonous graphs could be accumulated into larger panels (3-4). I think it will help the reader to perceive the work.
Figures' statistics also may be improved. For example, I have seen a few ways of displaying significance between entries recently, see Figure 3 from 'Stimulating photosynthetic processes increases productivity and water-use efficiency in the field' by Lopez et al. or see figure 3 from 'Simultaneous stimulation of sedoheptulose 1, 7‐bisphosphatase, fructose 1, 6‐bisphophate aldolase and the photorespiratory glycine' by the same group. Another option would be a table.
Under Figure 1, please explain "SPAD units".
Graphs of temperature changes and perspiration can also be cumulated. Unfortunately, there is no information on where the Authors took the data from - own measurements, the climate station? At what latitude and longitude? Authors should also put this information in the figure/panel description.
As the changes are quite big in terms of the data visualization itself, it seems to me that you can judge them as MAJOR. Nevertheless, the work is interesting, and I think it will interest the journal readers.
Author Response
Please see the attached file with our answers.

Reviewer 3 Report
Paragraph starting line 91- could be moved to M&M. Connecting lettrs in figure 2 seem to be incorrect. The same goes for Fig 3. pleae check the connecting letters for the Tukeys HSD, something seems to be off. Remove the grey box outline from the figures. Check the units in the figures on y axis, they are not the same for all graphs. the right way to write the units would be mg mL-1, not mg-1 mL-1. check this in the figures and throughout the text. Reference 22 - journal name in italics Reference 43 - shorten the name of journal and also italics. Check the whole reference list for inconsistencies!! Overall a nice manuscript with good discussion - please also check the English once more, there are a few mistakes within the text.Author Response
Please see the attached file.

Round 2
Reviewer 2 Report
I would like to thank the authors for complying with some of the comments.
He notices figure 3 - the x axis has disappeared along with the captions and there is no value on the y axis. The same in figures 4, 5, 6 and 7.
Please improve.
Author Response
The authors thank a lot for the reviewer's comment. Th figures are corrected.